# Experimental time-reversed adaptive Bell measurement towards all-photonic quantum repeaters

Yasushi Hasegawa[1], Rikizo Ikuta [1,2], Nobuyuki Matsuda [3,9], Kiyoshi Tamaki[4], Hoi-Kwong Lo[5,6,7], Takashi Yamamoto [1,2], Koji Azuma[3,8] & Nobuyuki Imoto[1,2]

An all-optical network is identified as a promising infrastructure for fast and energy-efficient communication. Recently, it has been shown that its quantum version based on 'all-photonic quantum repeaters'—inheriting, at least, the same advantages—expands its possibility to the quantum realm, that is, a global quantum internet with applications far beyond the conventional Internet. Here we report a proof-of-principle experiment for a key component for the all-photonic repeaters—called all-photonic time-reversed adaptive (TRA) Bell measurement, with a proposal for the implementation. In particular, our TRA measurement—based only on optical devices without any quantum memories and any quantum error correction—passively but selectively performs the Bell measurement only on single photons that have successfully survived their lossy travel over optical channels. In fact, our experiment shows that only the survived single-photon state is faithfully teleported without the disturbance from the other lost photons, as the theory predicts.

[1] Graduate School of Engineering Science, Osaka University, Toyonaka, Osaka 560-8531, Japan. [2] Quantum Information and Quantum Biology Division, Institute for Open and Transdisciplinary Research Initiatives, Osaka University, Osaka 560-8531, Japan. [3] NTT Basic Research Laboratories, NTT Corporation, 3-1 Morinosato Wakamiya, Atsugi, Kanagawa 243-0198, Japan. [4] Graduate School of Science and Engineering for Research, University of Toyama, Gofuku 3190, Toyama 930-8555, Japan. [5] Center for Quantum Information and Quantum Control (CQIQC), University of Toronto, Toronto, ON M5S 3G4, Canada. [6] Department of Physics, University of Toronto, 60 St. George St, Toronto, ON M5S 1A7, Canada. [7] The Edward S. Rogers Sr. Department of Electrical & Computer Engineering, University of Toronto, 10 King's College Road, Toronto, ON M5S 3G4, Canada. [8] NTT Research Center for Theoretical Quantum Physics, NTT Corporation, 3-1 Morinosato Wakamiya, Atsugi, Kanagawa 243-0198, Japan. [9] Present address: Department of Communications Engineering, Graduate School of Engineering, Tohoku University, Sendai 980-8579, Japan. Correspondence and requests for materials should be addressed to T.Y. (email: yamamoto@mp.es.osaka-u.ac.jp) or to K.A. (email: azuma.koji@lab.ntt.co.jp) or to N.I. (email: imoto@mp.es.osaka-u.ac.jp)

Quantum internet[1]—the quantum version of the current Internet—holds promise for accomplishing quantum teleportation[2], quantum key distribution (QKD)[3,4] and precise synchronisation of atomic clocks[5] among arbitrary clients all over the globe, as well as longer-baseline telescopes[6] and possibly even simulation of quantum many-body systems[1]. To realise it in a global scale for arbitrary users, it is reasonable to utilise not only recently developed satellites[7–10] but also existing optical networks that have already been installed in the world. An indispensable building block[11,12] for implementing such a quantum internet against photon loss of optical fibres is to use quantum repeaters over an optical network, irrespective of its topology[13–16]. For 17 years after the first proposal for quantum repeaters[17], it had widely been believed that their realisation needs demanding matter quantum memories[18,19] or matter qubits[20]. However, in 2015, this belief was disproved by a proposal of all-photonic quantum repeaters[21], which work without any matter quantum memories or matter qubits, that is, only with optical devices. Thanks to its all-optical nature, this scheme has advantages that cannot be seen in conventional quantum repeaters necessitating matter quantum memories. For instance, first, the repetition rate of the protocol could be as high as one wants, similarly to a memory-function-less scheme[20], as it is determined only by the repetition rate of the optical devices, independently of the communication distance. This would lead to a future higher-bandwidth quantum internet. Second, in principle, the scheme could work at room temperature and does not need any quantum interfaces among photons with different wavelengths, let alone between matter quantum memories and photons. Third, the scheme is an ultimate version of the all-optical network approach[22]—which has already been identified as a promising infrastructure for fast and energy-efficient communication in the field of conventional communication. Thus, the scheme is an important target of the development of photonic networks[23,24], and in particular their implementations using integrated quantum circuits[25–30] or frequency multiplexing[31,32]. The scheme would also represent a step towards a future all-photonic quantum computer[33–37].

In this Article, we report a proof-of-principle experiment for a key component of all-photonic quantum repeaters, which we will call all-photonic time-reversed adaptive (TRA) Bell measurement. In particular, the experiment has been implemented, based on our proposal of a scheme which requires an initial state composed of far fewer single photons than what needed[21,38] before. Besides, it does not need large-scale optical switches and quantum non-demolition measurement, let alone matter quantum memories and quantum error correcting codes. Our scheme is obtained by invoking the concept of the 'time-reversal' in the proposal of all-photonic quantum repeaters and by combining a local delayed preparation of a multipartite Greenberger–Horne–Zeilinger (GHZ) state with utilisation of a special feature of the type-II fusion gate[36]. Our experiment shows that, as the theory predicts, only the survived single-photon state is faithfully teleported without the disturbance from the other lost photons. In principle, once the GHZ state is treated in a lossless manner, our scheme could double the achievable distance of QKD, that is, it could have the same impact as the all-photonic intercity QKD[38].

## Results

**Basics of quantum repeaters.** Before introducing our scheme, we start by reviewing the efficiency enhancement of quantum repeaters brought by the adaptive Bell measurement—which performs the Bell measurement only on qubits that have successfully shared entanglement with distant nodes. To see this, it is instructive to begin by considering the simplest quantum repeater scheme which uses only a single repeater node C in the middle of Alice (A) and Bob (B) separated over a distance $L$. Here, the node C is connected to Alice and Bob by optical fibres with length $L/2$, whose transmittance is described by $\eta_{L/2} := e^{-L/(2L_{att})}$ for an attenuation length $L_{att}$. Note that, if Alice and Bob do not utilise the node C as a quantum repeater node like the Ekert scheme[4] or if they just run a direct quantum communication scheme between them like the Bennett–Brassard protocol[3], the communication efficiency inevitably scales as $\eta_L$[11,12]. In this case, about 400 km for the use of standard optical fibres is a practical distance limitation.

Let us first consider the mechanism of the simplest quantum repeater scheme using matter quantum memories (Fig. 1a), as a representative of conventional quantum repeater protocols. This protocol starts with $l$ entanglement generation processes between nodes AC and between nodes CB. In particular, in each entanglement generation process, say in the $i$th process ($i = 1$, 2, …, $l$), the node A (B) locally prepares an entangled state between a single-photon $\alpha_i$ ($\beta_i$) and her qubit $a_i$ (his qubit $b_i$), where the qubit $a_i$ ($b_i$) can be regarded as virtual for the case of application to QKD[21], and then tries to supply the qubit $a_i$ ($b_i$) with entanglement with a matter quantum memory $c_i$ ($c_{i+l}$) at the node C by exchanging the single photon through the optical fibre. In practice, this trial succeeds only probabilistically, because of imperfection of local operations and the photon loss in the optical fibres. This success probability $p_g$ is normally described as $p_g(L/2) = c_g \eta_{L/2}$ with an overall success probability $c_g$ of the local operations. This means that, if the nodes AC (CB) run the entanglement generation processes more than about $p_g^{-1}(L/2)$ simultaneously in parallel, they can share, at least, an entangled pair almost deterministically. Then, the node C pairs its own matter quantum memories that are entangled with adjacent nodes A and B, respectively, and it performs Bell measurement on these pairs to convert the entangled pairs between AC and CB into entangled pairs between AB. This Bell measurement—applied only to halves of successfully entangled pairs—can be called adaptive Bell measurement. By including general cases (such as the Duan–Lukin–Cirac–Zoller protocol[18]) where the Bell measurement may succeed only probabilistically, say with a probability $p_s$, we can conclude that Alice and Bob can obtain an entangled pair when nodes AC and CB run about $p_g^{-1}(L/2)p_s^{-1}$ entanglement generation processes on average (that is, when $\simeq p_g^{-1}(L/2)p_s^{-1}$).

On the other hand, if Alice and Bob perform the entanglement generation processes between them directly (without using the repeater node C), they need to run the processes about $p_g^{-1}(L)$ times to obtain an entangled pair between AB. Since $p_g^{-1}(L) > p_g^{-1}(L/2)p_s^{-1}$ holds for large $L$, even the simplest quantum repeater scheme utilising only the single node C could be more efficient than the direct entanglement generation between Alice and Bob. More precisely, compared with the direct entanglement generation, the simplest quantum repeater scheme enables Alice and Bob to double their communication distance without decreasing the communication efficiency. Hence, it already enables us to extend the achievable distance of fibre-based quantum communication up to about 800 km, which is useful for intercity backbone quantum links[38]. A more general quantum repeater scheme with $2^s - 1$ repeater nodes ($s = 1, 2, …$)[18,19] for accomplishing further scaling improvement can be regarded as one obtained by concatenating this simplest quantum repeater scheme with the help of quantum error correction or quantum memories with long coherence time.

The essence of the scaling improvement of the simplest quantum repeater scheme is the adaptive Bell measurement at the node C. In the above simplest quantum repeater scheme, this adaptive Bell measurement can be performed thanks to three

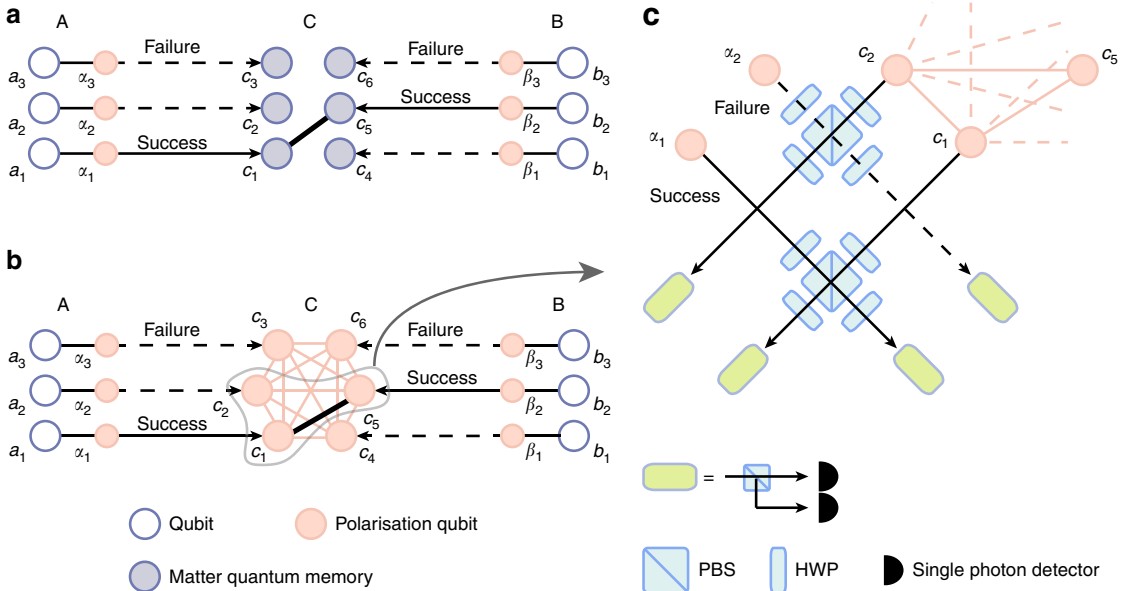

**Fig. 1** Adaptive Bell measurement for quantum repeaters. **a** Conventional quantum repeaters based on matter quantum memories are described, where Alice (A) or Bob (B) and the node C perform $l$ entanglement generation processes ($l = 3$ in the figure as an example). The essential of the quantum schemes is to perform the Bell measurement at the node C adaptively only on the pairs of matter quantum memories that have successfully shared entanglement with Alice and Bob through the entanglement generation processes. This adaptive Bell measurement is made possible thanks to the functionality of matter quantum memories. In contrast, our scheme described in **b** performs this adaptive Bell measurement without matter quantum memories, by invoking the conception of the 'time-reversal'. In particular, our scheme begins by entangling all the polarisation qubits $\{c_i\}_{i=1,2,...,2l}$ at the node C initially, followed by connecting/disentangling the qubits dependently on the success/failure of the entanglement generation processes. This switching between connecting and disentangling is passively performed by adopting the type-II fusion gates as the implementation for the Bell measurement between optical pulses $\alpha_i c_i$ and between $\beta_i c_{i+l}$, as depicted in **c**. The type-II fusion gate is composed of the polarising beam splitter (PBS), the half-wave plates and single-photon detectors

functions of the matter quantum memories $\{c_i\}_{i=1,2,...,2l}$ at the node C: (i) They can establish entanglement with a distant node A or B if the entanglement generation process succeeds. (ii) Their memory function enables the node C to keep entanglement until it is informed of the success/failure of the entanglement generation processes between AC and between CB. (iii) The independent accessibility to them enables the node C to selectively apply the Bell measurement only to halves of successfully entangled pairs, in order to avoid the error propagation from the other failed entanglement generation processes. Therefore, conventional schemes require the node C to be equipped with matter quantum memories. However, our all-photonic scheme shows that the adaptive Bell measurement is possible without using matter quantum memories.

**All-photonic time-reversed adaptive Bell measurement**. Our scheme—called TRA Bell measurement—is brought by invoking the concept of the 'time-reversal' in the proposal of all-photonic quantum repeaters[21]. In fact, this concept enables our scheme to work without matter quantum memories associated with three functions (i)–(iii). The conception of the time-reversal stems from the observation of the essential role of the adaptive Bell measurement at the node C, which is—among its own matter quantum memories $\{c_i\}_{i=1,2,...,2l}$—to entangle only local quantum memories that have successfully shared entanglement with distant nodes A and B. Unfortunately, this selective local supply of entanglement after the entanglement generation processes is impossible for our all-photonic scheme, because all the memories at the node C are replaced with single-photon qubits $\{c_i\}_{i=1,2,...,2l}$ without the function (ii) in our scheme. Instead, as shown in Fig. 1b, our scheme entangles *all*

the single-photon qubits $\{c_i\}_{i=1,2,...,2l}$ at the node C before the completion of entanglement generation processes. This means that our scheme essentially begins with entanglement swapping, rather than entanglement generation processes. In particular, the node C starts by locally preparing the single-photon qubits $\{c_i\}_{i=1,2,...,2l}$ in a $2l$-partite GHZ state[39]. Note that, up to the freedom of local unitary operators, this GHZ state is the same as the complete cluster state, corresponding to the state of the first-leaf qubits in the original proposal[21].

The next question for the node C in our scheme is how to perform the entanglement generation processes with the nodes A and B. This is possible by utilising a connection process for GHZ states: If we perform the Bell measurement between a qubit composing an $m$-partite GHZ state and a qubit composing another $n$-partite GHZ state, we obtain an $(m+n-2)$-partite GHZ state. To see how this works, we first note that the $i$th entanglement generation process between nodes AC (BC) starts with a local preparation of a bipartite GHZ state—equivalent to a Bell state—between the qubit $a_i$ ($b_i$) and a single-photon $\alpha_i$ ($\beta_i$). Hence, if the single-photon $\alpha_i$ from Alice ($\beta_i$ from Bob) successfully arrives at the node C and if the Bell measurement at the node C on these photons $\alpha_i c_i$ ($\beta_i c_{i+l}$) succeeds, an initial $m$-partite GHZ state ($m = 1, 2, ..., 2l$) having included the qubit $c_i$ ($c_{i+l}$) is transformed into an $m$-partite GHZ state which newly includes Alice's qubit $a_i$ (Bob's qubit $b_i$) instead of the qubit $c_i$ ($c_{i+l}$). This mechanism works as the function (i), that is, it works as the entanglement generation processes between nodes AC (BC). On the other hand, if the single-photon $\alpha_i$ ($\beta_i$) does not arrive at the node C—corresponding to the failure of the entanglement generation, we would like to disentangle the qubit $c_i$ ($c_{i+l}$) from the GHZ state, in order to satisfy function (iii) (or because such a qubit is associated with a matter quantum

memory that is not subjected to Bell measurement in the original simplest quantum repeater scheme). But, fortunately, we can disentangle the qubit $c_i$ ($c_{i+l}$) just by performing the $X$-basis measurement on it. Therefore, if the node C can switch between the Bell measurement and the $X$-basis measurement dependently on the arrival of single photons from Alice and Bob, we can perform the adaptive Bell measurement in a time-reversed manner, as well as the entanglement generation.

The final question for the node C is how to implement this switching between the Bell measurement and the $X$-basis measurement, by using only optical devices. For instance, such switching can be performed actively by combining quantum non-demolition (QND) measurement (based on an entangled photon source) and optical switches as in the all-photonic intercity QKD[38], or by using loss-tolerant encoding for qubits $\{c_i\}_{i=1,2,...,2l}$ as in the all-photonic quantum repeaters[21]. However, here we present the node C with another mechanism to perform the switching passively, rather than actively, in order to make it possible to start from far fewer single-photon resources. That is, in our case, the node C uses a delayed preparation of the $2l$-partite GHZ-state and the type-II fusion gate[36] (Fig. 1c) with photon-number-resolving detectors as the implementation of the Bell measurement on $\alpha_i c_i$ ($\beta_i c_{i+l}$). The delayed preparation means that the node C prepares the $2l$-partite GHZ state just before the arrival of optical pulses from Alice and Bob. This could enable us to assume that the GHZ state is lossless compared with the optical pulses sent from distant nodes AB (see Discussion for the precise requirement here). We also utilise a property of the type-II fusion gate as follows (see Supplementary Note 1): (a) If it is performed on the single-photon qubit $c_i$ ($c_{i+l}$) and the optical pulse $\alpha_i$ ($\beta_i$) in the vacuum state, it works as $X$-basis measurement on the qubit $c_i$ ($c_{i+l}$). (b) If it is performed on the single-photon qubit $c_i$ ($c_{i+l}$) and the optical pulse $\alpha_i$ ($\beta_i$) including a single photon, it works as either (b1) the $X$-basis measurement on the single photons or (b2) the Bell measurement. Therefore, in principle, our TRA Bell measurement scheme can be performed just by using the type-II fusion gates, without invoking techniques such as the QND measurement and the loss-tolerant encoding.

In this way, we have arrived at the following protocol: (1) For the entanglement generation processes, Alice and Bob locally prepare $l$ pairs of an optical pulse and a qubit in a Bell state (equivalent to a bipartite GHZ state). (2) Then, they send the $l$ optical pulses to the node C through the optical fibres simultaneously. (3) At the timing of the arrival of all the $2l$ optical pulses sent by Alice and Bob, the node C prepares $2l$ single-photon qubits in a $2l$-partite GHZ state locally and applies the type-II fusion gates to $2l$ pairs between the $2l$-partite GHZ state and the $2l$ arriving optical pulses. (4) Then, the node C announces the measurement outcomes. (5) If the measurement outcomes inform Alice and Bob of the existence of the successful entanglement generation processes between nodes AC and between nodes CB and every measurement outcome corresponds to the case (a), (b1) or (b2), the qubits of Alice and Bob should be in a GHZ state. Except for these events, the protocol is considered to fail. The size of the final GHZ state shared by Alice and Bob depends on how many successful entanglement generation processes have existed. Depending on this, individual bit-flip errors can be compensated thanks to the robustness of the GHZ state against such errors.

**Proof-of-principle experiment**. Let us now discuss about the experimental demonstration of our all-photonic TRA Bell measurement with a three-partite photonic GHZ state $|GHZ\rangle_{c_1 c_2 c_5}$ as described in Fig. 1c. In particular, we suppose a situation where

Alice sends two optical pulses $\alpha_1$ and $\alpha_2$ in single-photon states to the node C but only one of them survives the travel from the node A to the node C. Here, the node C does not know, in advance, which single photon is lost during the travel. Nevertheless, according to the theory above and described in Fig. 1c, just by performing the type-II fusion gates on $\alpha_1 c_1$ and on $\alpha_2 c_2$, the node C should share a Bell state between its remaining qubit $c_5$ and Alice's qubit $a_i$ which has been in a Bell state with the only survived single-photon $\alpha_i$ ($i = 1, 2$). In other words, arbitrary quantum states of only the survived single photon $\alpha_i$ should be teleported into the qubit $c_5$ without the disturbance from the other lost photon, because the state of the photon $\alpha_i$ can be chosen arbitrary by performing measurement on its partner $a_i$. Here, we experimentally demonstrate this quantum teleportation as the most primitive function of our TRA Bell measurement.

Our experimental setup is shown in Fig. 2. Pump light at 395 nm with a power of 600 mW for spontaneous parametric down-conversion (SPDC) is prepared by the frequency doubler with the light at 790 nm from a mode-locked Ti:sapphire (Ti:S) laser with a pulse width of 100 fs and a repetition rate of 80 MHz. The vertically ($V$-) polarised pump light is injected to a pair of type-I phase-matched and 1.5-mm-thick $\beta$-barium borate (BBO) crystals[40,41] to have a horizontally polarised photon pair $|HH\rangle_{\alpha\gamma_2}$. The subscripts indicate modes of photons. After passing through a quarter wave plate (QWP) with its axis oriented at 22.5° to $H$ polarisation, the pump light is reflected back to the QWP and the BBO crystals by a dichroic mirror to prepare a polarisation entangled photon pair $|\phi^+\rangle_{c_5 \gamma_1} = (|HH\rangle_{c_5 \gamma_1} + |VV\rangle_{c_5 \gamma_1})/\sqrt{2}$. By postselecting events where a single photon appears in mode $\alpha$, we obtain an $H$-polarised single photon in mode $\gamma_2$. The polarisation of photon $\gamma_2$ is rotated to diagonal polarisation by a half-wave plate (HWP), and then photons $\gamma_1$ and $\gamma_2$ are mixed at a polarising beam splitter (PBS$_{12}$). When two photons appear in output modes $c_1$ and $c_2$, the three-photon GHZ state $|GHZ\rangle_{c_1 c_2 c_5} = (|HHH\rangle_{c_1 c_2 c_5} + |VVV\rangle_{c_1 c_2 c_5})/\sqrt{2}$ is obtained[42–44].

For demonstrating the TRA Bell measurement, we use photon $\alpha$ not only as the heralding photon but also as Alice's signal photon, whose quantum state is encoded into $|\psi\rangle = \xi|H\rangle + \zeta|V\rangle$. As we have mentioned, in this experiment, we simulate a case where Alice sends two optical pulses in single-photon states to the node C but only one photon of them survives the lossy travel from the node A to the node C (Fig. 1b). For this, the signal photon in mode $\alpha$ is divided into two modes $\alpha_1$ and $\alpha_2$ by a 50:50 beam splitter (BS), corresponding to 50% loss of the transmission. If the signal photon is transmitted to mode $\alpha_1$ at the BS, photons $\alpha_1$ and $c_1$ interfere at PBS$_{11}$ for a type-II fusion gate (the lower side in Fig. 2), while photon $c_2$ alone goes to the other fusion gate (the upper side in Fig. 2). In this case, if detector $D_{21}$ is clicked, photon $c_2$ is projected onto $|D\rangle = (|H\rangle + |V\rangle)/\sqrt{2}$. Then, the GHZ state $|GHZ\rangle_{c_1 c_2 c_5}$ is projected onto a Bell state as $\langle D|_{c_2}|GHZ\rangle_{c_1 c_2 c_5} \propto |\phi^+\rangle_{c_1 c_5}$. As a result, by the type-II fusion gate on photons $\alpha_1$ and $c_1$, the state of photon $\alpha_1$ is teleported to the state of photon $c_5$ as $\langle \phi^+|_{\alpha_1 c_1}|\psi\rangle_{\alpha_1}|\phi^+\rangle_{c_1 c_5} \propto |\psi\rangle_{c_5}$. On the other hand, when detector $D_{22}$ is clicked, photon $c_5$ is projected onto $|A\rangle = (|H\rangle - |V\rangle)/\sqrt{2}$. Then, the GHZ state is projected onto $\langle A|_{c_2}|GHZ\rangle_{c_1 c_2 c_5} \propto |\phi^-\rangle_{c_1 c_5}$, where $|\phi^-\rangle_{c_1 c_5} \equiv (|HH\rangle_{c_1 c_5} - |VV\rangle_{c_1 c_5})/\sqrt{2}$. After the fusion operation on photons $\alpha_1$ and $c_1$, the state of photon $c_5$ becomes $\langle \phi^+|_{\alpha_1 c_1}|\psi\rangle_{\alpha_1}|\phi^-\rangle_{c_1 c_5} \propto \xi|H\rangle_{c_5} - \zeta|V\rangle_{c_5}$. The phase shift here can be compensated by applying the phase-flip operation, and thus we can obtain $|\psi\rangle_{c_5}$. In our proof-of-principle experiment, we skip this feed-forward phase-flip operation for simplicity.

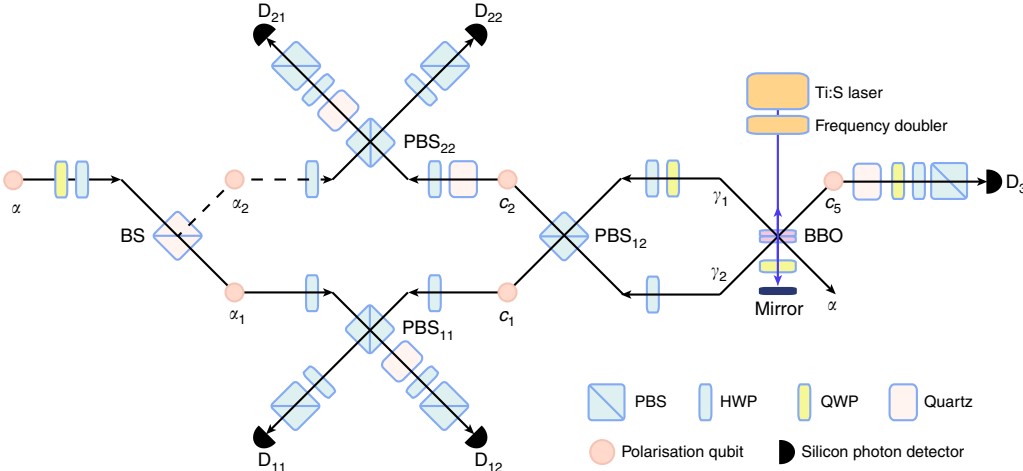

**Fig. 2** Experimental setup for our all-photonic TRA Bell measurement. This setup corresponds to the measurement in Fig. 1c. Two input modes $\alpha_1$ and $\alpha_2$ are prepared by a 50:50 beam splitter (BS) with the incoming mode $\alpha$ that comes from one output mode of the SPDC at BBO. The single photon in mode $\alpha$ travels into $\alpha_1$ or $\alpha_2$, which simulates the lossy quantum channel from Alice to Bob. The rest of the photons in modes $c_1$, $c_2$ and $c_5$ generated by the SPDC at BBO forms the tripartite GHZ state necessary for TRA Bell measurement at node C. The lower and the upper type-II fusion gates tell the success/failure of the arrival of the photon from $\alpha_1$ or $\alpha_2$ and of the teleportation of its state to the photon in mode $c_5$. The teleported state is estimated by controlling the quater wave plate (QWP), the half wave plate (HWP) and the polarising beam splitter (PBS). Quartz plates are used for adjusting additional phase shifts. All silicon photon detectors are coupled to single-mode optical fibres

Similarly, when the signal photon $\alpha$ is reflected to mode $\alpha_2$ at the BS, photon $\alpha_2$ and photon $c_2$ interfere at the fusion gate in the upper side and only photon $c_1$ goes to the fusion gate in the lower side. As a result, if detector $D_{11}$ is clicked, the final state in mode $c_5$ becomes $|\psi\rangle_{c_5}$. If detector $D_{12}$ is clicked, the state in mode $c_5$ becomes $\xi|H\rangle_{c_5} - \zeta|V\rangle_{c_5}$.

In our experiment, photon $\alpha$ is mixed with a photon either in mode $c_1$ or $c_2$, and, at the same time, photon $\alpha$ is used for heralding the successful generation of the GHZ state among modes $c_1$, $c_2$ and $c_5$ at the fusion gate. Here we explain how it works properly in our experiment. When the perfectly mode-matched two photons in modes $\gamma_1$ and $\gamma_2$ interfere at the PBS$_{12}$ for generating the GHZ state, they come out from modes $c_1$ and $c_2$ with probability 1/2, or both of the two photons come out together from the same mode $c_1$ or $c_2$ with probability 1/4. The former case is a desired event as we described before. Apparently, the latter case may cause the degradation of the quality of the experiment. However, when the two photons are bunched in mode $c_1$ or $c_2$ after PBS$_{12}$, the two-photon state always becomes the NOON state of diagonal and anti-diagonal polarisation which are rotated to $H$ and $V$ polarisation at the fusion gate. As a result, both of the two photons are transmitted or reflected at the PBS in the fusion gate, and thus such events never contaminate the successful coincidence events among three of detectors $D_{11}$, $D_{12}$, $D_{21}$ and $D_{22}$. In order to obtain well mode matching of photons $\gamma_1$ and $\gamma_2$, we adjust the temporal mode by mirrors on the motorised stage, select the frequency mode of the photons by an interference filter with a bandwidth of 2.7 nm, and use the single-mode fibre connected to the avalanche photon detectors (quantum efficiency is about 60%). Furthermore, for suppressing the multiple photon emission by the SPDC, we set the photon pair generation probability per pulse at the BBO crystals to $\sim 5 \times 10^{-3}$. We also notice that BS, PBS$_{11}$ and PBS$_{22}$ add $\pi$ phase shift between $|H\rangle$ and $|V\rangle$, resulting in the $\pi$ phase shift of the teleported state in some coincidence patterns. Again, although this can be compensated by the feed-forward phase-flip operation, in the following experiment, we read the measurement basis state as one subjected to a proper phase-flip unitary, instead of the feed-forward operation, for simplicity.

In the experiment, we first performed quantum state tomography[45] of the photon pair in modes $c_5$ and $\gamma_1$ prepared by the SPDC. The photons were detected at $D_3$ and $D_{11}$. We reconstructed the density operator $\rho_{c_5\gamma_1}$ of the two-photon state by using the iterative maximum likelihood method[46]. The observed fidelity $\langle\phi^+|\rho_{c_5\gamma_1}|\phi^+\rangle$ was 0.95(02), where digits in parentheses represent the standard deviation, for example, $0.95 \pm 0.02$. Matrix representation of $\rho_{c_5\gamma_1}$ and the observed coincidence counts are shown in Fig. 3. The count rate was about 2500 Hz.

Next we performed the quantum state tomography of the three-photon state in modes $c_5$, $c_2$ and $c_1$ prepared by using $\rho_{c_5\gamma_1}$ and the photon in mode 2 heralded by the detection of photon $\alpha$ by detector $D_{22}$. For this measurement, we did not rotate the HWPs just before and after the PBSs at the fusion gates. The tomography was performed by using detectors $D_{11}$, $D_{21}$ and $D_3$. The reconstructed density operator $\rho_{c_5c_2c_1}$ is shown in Fig. 4. The observed fidelity $F_{GHZ} = \langle GHZ|\rho_{c_5c_2c_1}|GHZ\rangle$ was 0.84(05) with a count rate of about 0.06 Hz. With the witness operator defined by $\mathcal{W} = 3I/4 - |GHZ\rangle\langle GHZ|$[47], $\mathrm{Tr}(\rho_{c_5c_2c_1}\mathcal{W}) = -0.09(05) < 0$ was confirmed, which clearly shows that the obtained state was in the genuine GHZ class.

Finally, we demonstrated the TRA Bell measurement. We prepared four states $|H\rangle$, $|V\rangle$, $|D\rangle = (|H\rangle + |V\rangle)/\sqrt{2}$ and $|L\rangle = (|H\rangle - i|V\rangle)/\sqrt{2}$ for $|\psi\rangle$ in mode $\alpha$. We performed the quantum state tomography of the teleported state in mode $c_5$ and reconstructed its density operator for the four input states conditioned by the successful events of the two fusion gates. We list the experimental results in Table 1. From Table 1, we see that the teleportation succeeded with high fidelities for the four input states and for any detection events. Although the fidelities relatively deviate from each other due to the statistical fluctuations caused by the low-count events, we can confirm that the teleportation succeeded with high fidelities for most of the four input states and for any detection events.

By using the above observed counts, we reconstructed the process matrices $\chi$ of the teleportation channels from input photon $\alpha$ to output photon $c_5$ by the technique of the process

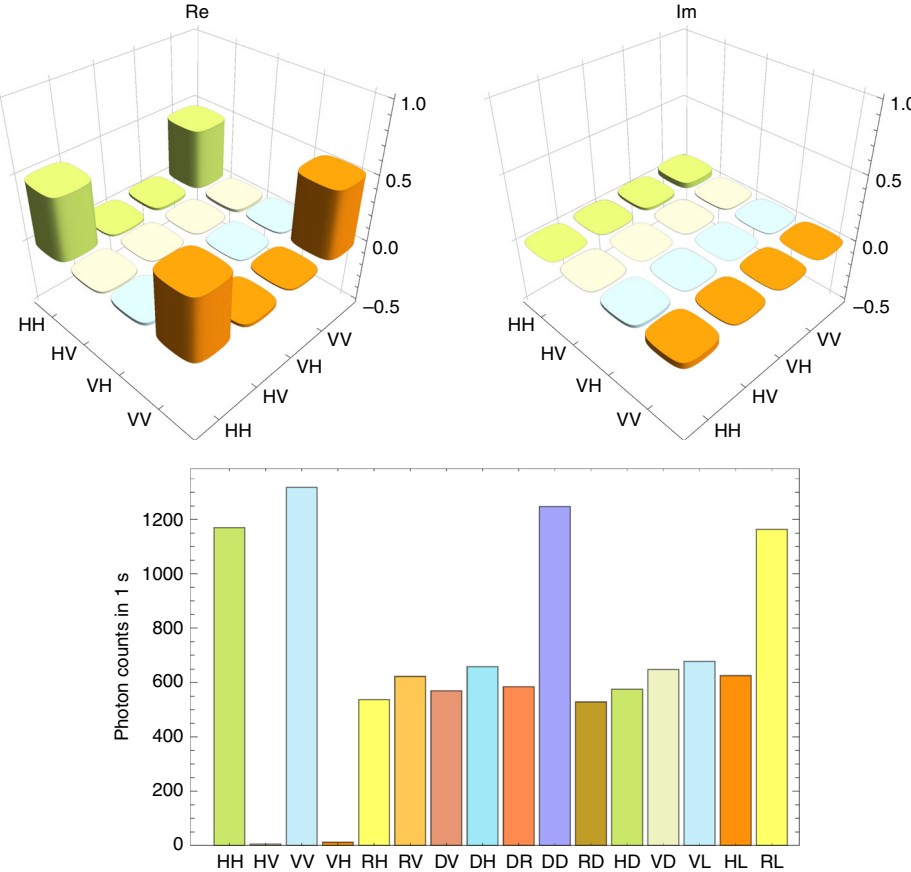

**Fig. 3** Estimated quantum states expected to be a Bell state. The upper graphs show the real and imaginary part of the reconstructed density matrix $\rho_{c_5\gamma_1}$. The lower graph shows the measured coincidence counts in 16 basis states necessary for the reconstruction of the density matrix

tomography[48]. The process matrix is calculated by the input–output relation $\rho_{\text{out}} = \sum_{m,n=0}^{3} \sigma_m \rho_{\text{in}} \sigma_n \chi_{mn}$, where $\rho_{\text{in}}$ ($\rho_{\text{out}}$), $\sigma_i$ and $\sigma_0$ represent the input (output) density matrix, the Pauli matrices and the identical matrix, respectively. The ideal process matrix $\chi^{\text{ideal}}$ of the teleportation has only one non-zero element $\chi_{00}^{\text{ideal}} = 1$. The result of the reconstruction is shown in Fig. 5. The obtained process fidelity $F_{\text{p}} \equiv \text{Tr}(\chi\chi^{\text{ideal}})$ is given in Table 1. From these results, we conclude that our scheme works as quantum teleportation with two standard deviations apart from the classical limit 0.5 of the process fidelity[49], irrespectively of whether Alice's signal photon $\alpha$ passes through either $\alpha_1$ or $\alpha_2$. The degradation of the process fidelity is mainly caused by the mode mismatch between relevant photons. Especially, it seems that our complex interferometer with GHZ state generation and type-II fusion gates further increases the mode mismatch due to the misalignments. For example, the visibility of 0.82 for two-photon interference at GHZ state generation is relatively lower than typical value 0.90 obtained in a simpler experimental setup[50]. Integrated photonic circuits and indistinguishable photon sources, which have been actively studied recently[25–30,51], could give a solution to this issue in the near future.

### Discussion

We have proposed an all-photonic TRA Bell measurement and experimentally confirmed the working principle of the adaptive Bell measurement necessitated by all-photonic intercity QKD and all-photonic quantum repeaters. Although our scheme is designed mainly to experimentally demonstrate this proof-of-principle experiment, one might be interested in the potential of our current scheme. Here, we discuss this.

To check the potential of our protocol, among various types of error and imperfection, we consider the effect of non-unit quantum efficiency $1 - \epsilon_0$ of photon-number-resolving detectors as dominant[21,52] and inevitable noise for our protocol. Assuming the use of these imperfect detectors, we plot the asymptotic final secret key rate $G := \max_l G_l$ of our protocol in Fig. 6, where $G_l$ represents average secret bits per the number $l$ of multiplexing (see the derivation in Supplementary Note 2 and optimal choices of $l$ in Supplementary Table 1 for several cases). From the figure, we can confirm that our protocol has a scaling of $\eta_{L/2}$ in the ideal case of $\epsilon_0 = 0$ as the theory suggests. This is good in the sense that our protocol has no scaling gap with the private capacity[14–16] (the upper dashed curve in the figure) for the optical network of Alice, Bob and the middle node C—that is, the ultimate performance which can be achieved when Alice, Bob and the node C are allowed to use ideal universal quantum computers freely. However, the figure also suggests that the performance is very sensitive to the quantum efficiency. This is because, in the case of $\epsilon_0 > 0$, even if a type-II fusion gate at the node C finds the arrival of a single photon, there might be the case where the found single photon has come from Alice or Bob, rather than the locally prepared $2l$-partite GHZ state, leading to the phase error of the final GHZ state of Alice and Bob. In short, in principle, our protocol could achieve almost best performance allowed by quantum mechanics, but, in practice, it necessitates very high quantum efficiency.

Note that our protocol can be combined with loss-tolerant encoding. Indeed, even if a good photon detector with high quantum efficiency is unavailable, our 'delayed preparation' idea is still important to perform the time-reversed adaptive Bell

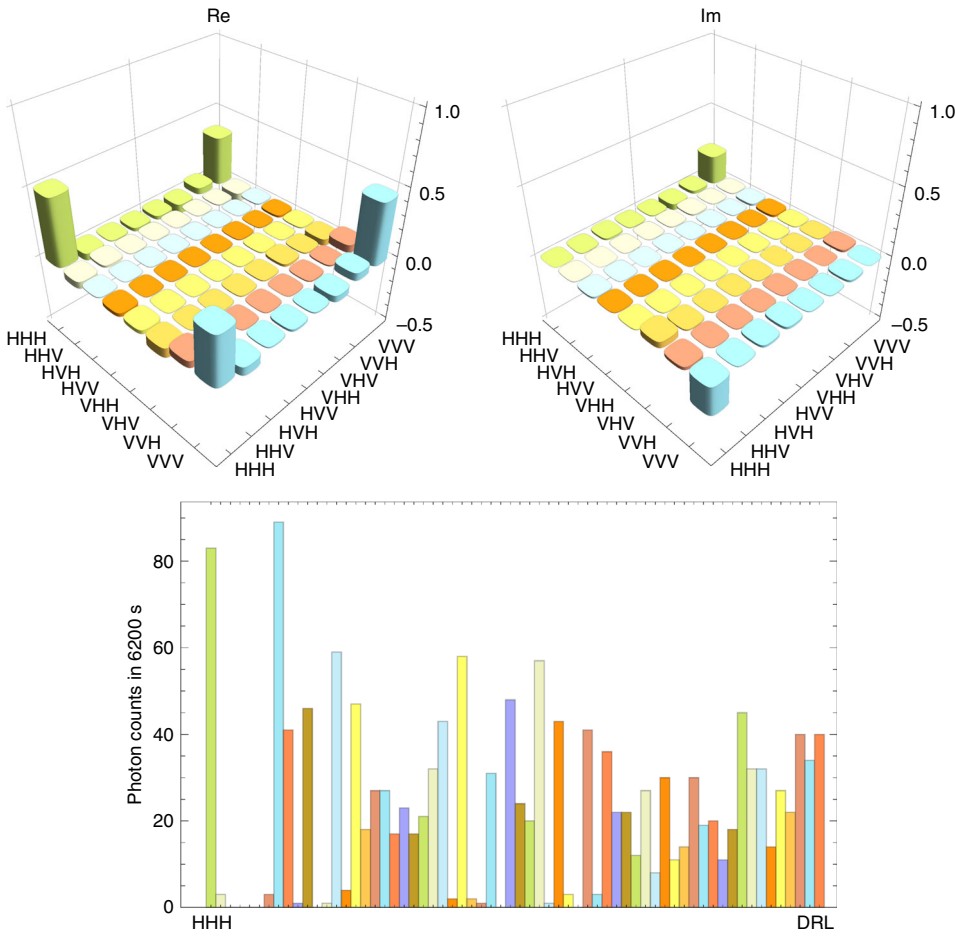

**Fig. 4** Estimated quantum states expected to be a tripartite GHZ state. The upper graphs show the real and imaginary part of the reconstructed density matrix $\rho_{c_5c_2c_1}$. The lower graph shows the measured coincidence counts in 64 basis states necessary for the reconstruction of the density matrix

| Table 1 Fidelities of the output states to the ideal states and the process fidelities to the ideal teleportation process | | | | | | | |
|---|---|---|---|---|---|---|---|
| $D_{21}$ and $D_{22}$ and $D_{11}$ | | $D_{21}$ and $D_{22}$ and $D_{12}$ | | $D_{21}$ and $D_{11}$ and $D_{12}$ | | $D_{22}$ and $D_{11}$ and $D_{12}$ | |
| Desired state | Fidelity | Desired state | Fidelity | Desired state | Fidelity | Desired state | Fidelity |
| $|H\rangle$ | 0.71(10) | $|H\rangle$ | 0.93(05) | $|H\rangle$ | 1.00(09) | $|H\rangle$ | 0.90(07) |
| $|V\rangle$ | 0.96(04) | $|V\rangle$ | 0.93(06) | $|V\rangle$ | 0.77(11) | $|V\rangle$ | 0.95(05) |
| $|D\rangle$ | 0.83(11) | $|D\rangle$ | 0.73(07) | $|D\rangle$ | 0.56(11) | $|D\rangle$ | 0.70(13) |
| $|L\rangle$ | 0.70(06) | $|L\rangle$ | 0.71(10) | $|L\rangle$ | 0.79(11) | $|L\rangle$ | 0.81(06) |
| $F_p = 0.80(08)$ | | $F_p = 0.73(08)$ | | $F_p = 0.90(10)$ | | $F_p = 0.73(09)$ | |

The output states in mode $c_5$ are estimated by detectors $D_{11}$, $D_{12}$, $D_{21}$ and $D_{22}$ at the fusion gates. When the input state in mode $\alpha$ is $|H\rangle$ and $|V\rangle$, the measurement time was 19,800 s for each basis setting. That was 59,600 s for the input state of $|D\rangle$ and $|L\rangle$. For the reconstruction of the output states in mode $c_5$, we removed the fourfold coincidence events among $D_{11}$, $D_{12}$, $D_{21}$ and $D_{22}$ as error events. The method of the estimation of the process fidelity $F_p$ is written in the main text. The digits in parentheses represent the standard deviations calculated by the observed counts with an assumption of the Poisson statistics

measurement, with far fewer photons than the original all-photonic quantum repeater protocol[21], via loss-tolerant encoding. In particular, even if the type-II fusion gates do not work properly to disentangle unnecessary qubits of the $2l$-partite GHZ state due to $\epsilon_0 > 0$, the best scaling of $\eta_{L/2}$ can still be obtained by modifying our protocol such that it uses an encoded complete-like cluster state $\left|\bar{G}_c^l\right\rangle$ appearing in the original protocol[21], rather than the $2l$-partite GHZ state. This is because, in this case, if necessary, the first-leaf qubits of the state $\left|\bar{G}_c^l\right\rangle$—associated with qubits in the $2l$-partite GHZ state conceptually—are encoded to be able to be disentangled faithfully and almost deterministically even if photons receive not only loss corresponding to the effect of $\epsilon_0 > 0$

but also depolarisation errors[21]. Nevertheless, the required number of photons to compose the first-leaf qubits here is far fewer than the original protocol. This is because our delayed preparation merely requires the first-leaf qubits to be robust against only local loss and only local depolarisation errors, rather than those occurring in their long travel to the adjacent repeater nodes, in contrast to the original protocol[21]. Therefore, even in the case of $\epsilon_0 > 0$, our delayed preparation idea enables our protocol to be modified so as to achieve the best scaling of $\eta_{L/2}$.

As have been shown here, depending on the efficiency of photon detectors and the size of the multipartite entangled states that one can prepare at node C, our scheme will gradually expand

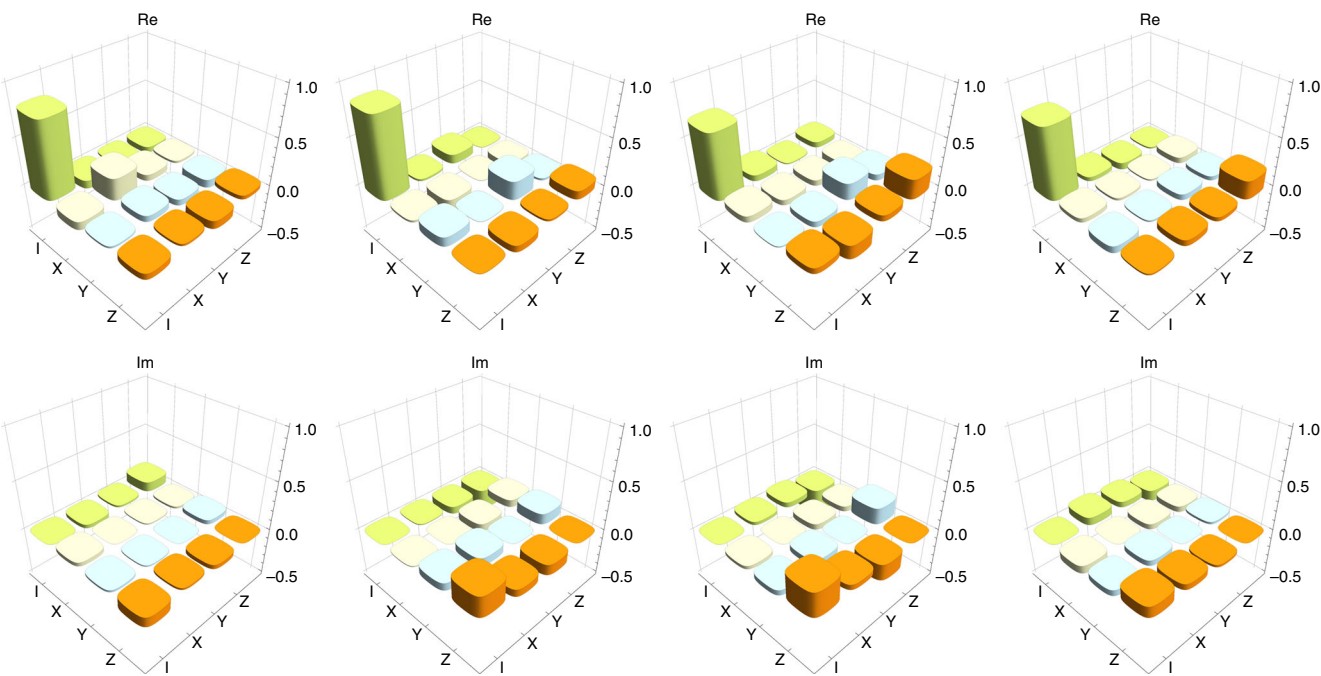

**Fig. 5** The reconstructed process matrices. From the left, the matrices are estimated in the cases of the coincidence detections among $D_{21}$ and $D_{22}$ and $D_{11}$, $D_{21}$ and $D_{22}$ and $D_{12}$, $D_{21}$ and $D_{11}$ and $D_{12}$, and $D_{22}$ and $D_{11}$ and $D_{12}$ from the left. Upper and lower graphs show the real and imaginary part of the process matrixes. All process matrices have a large element corresponds to the identity operation, showing high fidelities of the performed quantum teleportation, as listed in Table 1

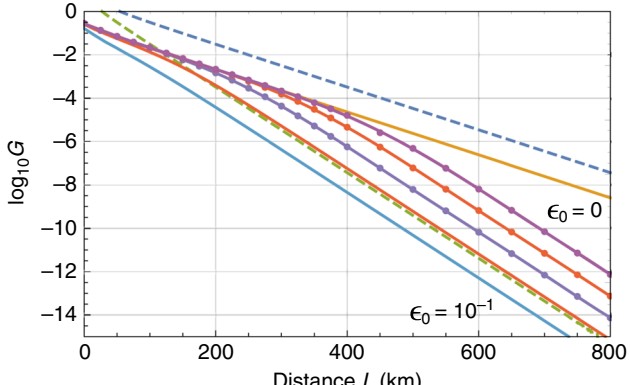

**Fig. 6** Performance of our protocol. We assume photon-number-resolving detectors with quantum efficiency $1 - \epsilon_0$. We assume $L_{att} = 22$ km. Solid curves represent the asymptotic final secret key rate $G$ (per pulse) for $\epsilon_0 = 0, 10^{-5}, 10^{-4}, 10^{-3}, 10^{-2}, 10^{-1}$ from the above in order, while the upper and lower dashed curves represent the private capacities of the optical network[14–16] of Alice, Bob and the middle node C and of point-to-point communication[12] between Alice and Bob, respectively. Solid curves with dots are obtained by fitting a smaller number of dots, compared with other curves. From this figure, we can confirm that our protocol has a scaling of $\eta_{L/2}$ in the ideal case of $\epsilon_0 = 0$, with no scaling gap with the private capacity. We also see that our protocol is quite sensitive to local loss and requires very high detection efficiency. However, as explained in the main text, our protocol could be further improved via loss-tolerant encoding

the achievable distance of the QKD up to 800 km, like all-photonic intercity QKD[38]. Subsequently, TRA Bell measurement with matured loss-tolerant encoding[53,54] and a good single/entangled photon source or a cluster-state machine gun[51,55,56] contributes the realisation of all-photonic quantum repeaters or internet. The

concept of TRA Bell measurement can also be applied to matter systems such as trapped ions[57,58] and superconducting qubits[59] in order to achieve a distributed quantum information processing architecture, including quantum repeaters.

## Data availability
The data that support the findings of this study are available from the corresponding authors on request.

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

## Acknowledgements

This work was supported by CREST, JST JPMJCR1671; MEXT/JSPS KAKENHI Grant number JP16H02214, JP18H04291, JP15KK0164 and JP16K17772; NSERC, US Office of Naval Research, Huawei Technologies Canada Co., Ltd., CFI and ORF.

## Author contributions

K.A., N.M., R.I. and T.Y. planed the research. R.I. and Y.H. designed the experiment. Y.H. and R.I. carried out the experiments and analysed the data under supervision of T.Y. and N.I. K.A. organised the theoretical part with K.T. and H.L. All authors contributed to the discussions and interpretations. K.A., R.I. and T.Y. wrote the manuscript with input from all authors.

## Additional information

**Competing interests:** The authors declare no competing interests.

