## [Peer Review File · Nature Communications]

Reviewers' comments:

Reviewer #1 (Remarks to the Author):

The paper reports an experimental attempt at demonstrating the type of measurements we may need in certain storage-free quantum repeaters. In particular, they generate a GHZ state and interact it with a single photon to mimic what they call an adaptive Bell measurement. The paper as a whole is well written (although some additional proof reading is needed), and it is easy to understand. I have one serious reservation, however, on the twist that the authors give to the applicability of this to quantum repeaters. My reservation is on two issues:

1) if the system they propose is scalable or beats known no-repeater bounds? In the latter case, what requirements are there for the employed devices?

2) if this experiment gets ever close to such requirements.

My handwavy calculations suggest that in order that the suggested protocol (with one cluster state in the middle) beats the PLOB bound in [12] we need extremely high detector efficiencies and a large cluster state in the middle. This is even without considering the low generation rate for the cluster state, which could even impose serious challenges on the all-photonic quantum repeater proposal. This also seems less favorable to the proposal in [36], which does not impose a restriction on the detector efficiency but requires fast low-loss switching. With regard to my second concern the experiment does not even get close to what may be eventually needed neither in terms of rate nor fidelity.

I understand that this is a proof of principle experiment, but the chosen title and the twist that the authors give in the introduction and the early parts of their paper may create the impression that we have a breakthrough in repeater technologies, whereas from what I understand this is another experiment that explores how we can effectively do a Bell measurement.

Some other minor issues:

- We have terms like memory-less versus all-photonic quantum repeaters. To me, the common feature is that they do not necessarily need to store quantum states, whereas for quantum processing or generation of certain complicated quantum states they may still need to use quantum memories. The first storage-free quantum repeater proposal I believe has appeared here: Nature Photonics volume 6, pages 777–781 (2012) doi: 10.1038/nphoton.2012.243 but, there is no reference to this work.

- The "adaptive" Bell measurement term used in here and in [36] I believe has predecessors of "informed" or "acknowledged" Bell measurement. Some reference must be given to previous work on this and why the authors choose to use a different term here.

- I'm not sure the references made to all-optical classical communications provide a good analogy between an all-photonic quantum repeater and all-optical networks. In the latter the focus is to avoid doing any measurements on the travelling light and just switch light beams around until they get to the destination. In the quantum version, we will have lots of measurement, so not sure about this pitch either.

So, overall, I think there might be some novelty in the way the experiment has been designed (following some underpinning theoretical investigation), but I am not sure if it has been presented to that effect. I believe we are still far away from implementing all-photonic repeaters if we account for what is technologically needed for such systems to offer any advantage. What we need versus what we have got has not been addressed in the paper.

Reviewer #2 (Remarks to the Author):

In this submission the authors report on the implementation of a key ingredient for all-photonic quantum repeaters. The implementation comes together with a new theoretical proposal for Bell measurements. This improvement apparently reduces significantly the resources necessary to implement an all-photonic quantum repeater. This is very exciting, since the first proposals for an all-photonic quantum repeater seemed out of near future experimental reach or even impractical. However, the paper fails short to its title and abstract. There is no experimental all-photonic quantum repeater implemented as the title suggests, and the Bell measurement performed is only half of the newly introduced time-reversed adaptive (TRA) Bell measurement: in the experiment there is only Alice attached to the repeater station. Additionally, I miss a more detailed explanation of this TRA measurement, see comments below. With a more detailed description and accurate claims, I would support the acceptance of this exciting progress towards experimental all-photonic quantum repeaters.

- The title and abstract should be revised to reflect the content. This is a paper about experimental progress towards a key element in all photonic quantum repeaters, the TRA, but with only two parties that are not separated, the experiment cannot be called a repeater.
- Apparently this TRA "needs far fewer single photons than existing all-photonic schemes" (page 1). This claim is never substantiated, the authors should either support it with some table, figure, derivation, or drop it. More generally, I miss a comparison with previous proposals in section "All-photonic time-reversed adaptive bell measurement"
- A minor detail. In section "essence of quantum repeaters" there is a discussion using matter memories, however in the second column of page 2 it is said "by considering the fact that the bell measurement with linear optics succeeds only probabilistically". I am unsure I understood the setup, if we were talking about matter memories, why would we need a linear optics bell measurement?
- Second column of page 2 "More precisely, the simplest quantum repeater schemes doubles the communication distance without decreasing the communication efficiency". What does efficiency mean here?
- In page 7, it is claimed that "teleportation succeeded with high fidelities for the four input states and for any detection events". However, in table 1, one of the entries has fidelity 0.56 ± 0.11 . Two comments: 1) the claim should be moderated to include this entry and 2) I miss some discussion about this anomalous entry in the table.

Reviewer #3 (Remarks to the Author):

The manuscript by Hasegawa et al. describes a proof-of-principle experiment for a component of all-photonic quantum repeaters - the time-reversed adaptive Bell state measurement. The research supplements ongoing efforts in all photonic quantum networks and represents experimental research at the forefront of the field. I believe that the article can be of interest to a general science audience and thus support publication in Nature Communications once the following points have been addressed:

-- The results of the proof of concept experiment supersede the classical teleportation fidelity bound by only two s.d. statistical significance, and I believe that a more detailed discussion would be in order. What is the main experimental reason for this limited violation? What are the main challenges/critical technology requirements towards real applications? How does this compare to other approaches for (all-optical) quantum repeaters?

--In the discussion, the authors state that: "Depending on the efficiency and the size of the GHZ state ... our scheme will gradually expand the achievable distance of the QKD up to 800 km...".

The authors should provide quantitative statements in support of this claim, in particular regarding; (i) efficiency; (ii) number of parallel communication channels; (iii) cluster size; (iv) process fidelities, that are required for different link-loss scenarios.

--While I find the section "essence of quantum repeaters" very illustrative, I believe that the section "All-photonic time-reversed adaptive Bell measurement" should put the new results into clearer context with some of the co-authors previous work (in particular Ref. 19). At present the only link is the reference to the equivalence of the GHZ state in the repeater node to the "...1-st leaf qubits in the original proposal [19]".

--For someone outside of the field of optical quantum computation, the action of the type-II fusion gate, which plays a key role in the experimental scheme, might be difficult to grasp. I would encourage the authors to add a section describing (in more general terms) the main features, limitations, and its relevance to the overall scheme. This would make the article more accessible to a wider audience, in particular to engineers in the QKD community.

Also, in the description of the general scheme, the distinction between the polarization qubit and local qubit (Fig 1) seems a little at odds with the "all-photonic" approach. I think this potential source of confusion could be resolved if an intermediate sender node between A(B) & C were mentioned somewhere in the text.

Dear referees,

Thank you all for taking time to read our manuscript and for giving us your constructive comments. With those, our manuscript, specifically the Discussion section, is widely revised. It is our great pleasure if you read that Discussion part.

All the revision parts in this round are marked in the PDF file provided as a Related Manuscript File. If you need, please use it to know which parts are modified in this round. Besides, please notice that our paper now includes Supplementary Information (composed of Supplementary Notes 1 and 2) as well.

Dear Referee 1,

We sincerely thank Referee 1 for reviewing our manuscript and giving us a report. Your constructive comments (coloured blue in what follows) lead to a significant improvement of our paper, as can be confirmed below and by the revised manuscript. In what follows, we would like to reply to your comments in order. Note that the reference numbers appearing below represent ones in the revised manuscript.

The paper reports an experimental attempt at demonstrating the type of measurements we may need in certain storage-free quantum repeaters. In particular, they generate a GHZ state and interact it with a single photon to mimic what they call an adaptive Bell measurement. The paper as a whole is well written (although some additional proof reading is needed), and it is easy to understand.

Thank you for stating that our paper as a whole is well written and it is easy to understand.

I have one serious reservation, however, on the twist that the authors give to the applicability of this to quantum repeaters. My reservation is on two issues:

1) if the system they propose is scalable or beats known no-repeater bounds? In the latter case, what requirements are there for the employed devices?

2) if this experiment gets ever close to such requirements.

My handwavy calculations suggest that in order that the suggested protocol (with one cluster state in the middle) beats the PLOB bound in [12] we need extremely high detector efficiencies and a large cluster state in the middle. This is even without considering the low generation rate for the cluster state, which could even impose serious challenges on the all-photonic quantum repeater proposal. This also seems less favorable to the proposal in [36], which does not impose a restriction on the detector efficiency but requires fast low-loss switching. With regard to my second concern the experiment does not even get close to what may be eventually needed neither in terms of rate nor fidelity.

I understand that this is a proof of principle experiment, but the chosen title and the twist that the authors give in the introduction and the early parts of their paper may create the impression that we have a breakthrough in repeater technologies, whereas from what I understand this is another experiment that explores how we can effectively do a Bell measurement.

As you have already noticed, our scheme was originally designed for the experimental demonstration to confirm the working principle of the adaptive Bell measurement necessitated by all-photonic intercity QKD [38] and all-photonic quantum repeaters [21]. However, we thank you for pointing out a possible weakness of our protocol when we use photon detectors with non-unity quantum efficiency.

According to your comment, we performed a detailed theoretical analysis on the performance of our protocol. As a result, we found that, in the case of photon detectors with unity quantum efficiency, the

performance of our protocol has no scaling gap even with the private capacity of the optical network with a single middle repeater node, as the theory suggests. We also see that our protocol is quite sensitive to local loss and requires very high detection efficiency of photon detectors, as the referee has imagined. But, this problem can be overcome by using loss-tolerant encoding with far fewer single photons than original all-photon quantum repeater proposal [21].

The details of these are now summarized in the Discussion. The current Discussion section includes a figure of the performance of our protocol (which is based on a detailed calculation in newly added Supplementary Notes 1 and 2), and discussions noted above. Please read again, at least, the Discussion section.

In addition, please note that we now change the title into a more accurate one. Please also check this.

Some other minor issues:

- We have terms like memory-less versus all-photon quantum repeaters. To me, the common feature is that they do not necessarily need to store quantum states, whereas for quantum processing or generation of certain complicated quantum states they may still need to use quantum memories. The first storage-free quantum repeater proposal I believe has appeared here: Nature Photonics volume 6, pages 777–781 (2012)

doi:10.1038/nphoton.2012.243

but, there is no reference to this work.

We now refer to that paper as a proposal [20] that still uses matter qubits but does not require its memory function. Thank you for pointing out that reference.

- The "adaptive" Bell measurement term used in here and in [36] I believe has predecessors of "informed" or "acknowledged" Bell measurement. Some reference must be given to previous work on this and why the authors choose to use a different term here.

Sorry. So far, we have not yet been able to find a proper reference which uses “informed” Bell measurement or “acknowledged” Bell measurement. If you know it, could you kindly inform us of it?

- I'm not sure the references made to all-optical classical communications provide a good analogy between an all-photon quantum repeater and all-optical networks. In the latter the focus is to avoid doing any measurements on the travelling light and just switch light beams around until they get to the destination. In the quantum version, we will have lots of measurement, so not sure about this pitch either.

From a theoretical point of view, the ultimate goal of conventional all-optical networks is to encode a classical signal into optical pulses and to then send it only via optical devices, without changing its media. Analogously, the ultimate goal of all-photon quantum repeaters is to encode a quantum signal into optical pulses and to then send it only via optical devices, without changing its media. Although the

all-photonic quantum repeaters include many measurements as the referee states, the measurement outcome does not have any *quantum* correlation with the quantum signal (although it has classical correlation like classical outcomes in the quantum teleportation protocol), because, otherwise, such measurement should disturb the quantum signal from the no-cloning theorem (but this is not the case for all-photonic quantum repeaters). This means that quantum information is always embedded into optical pulses in the all-photonic repeater protocol, and this fact suggests that the concept of all-photonic quantum repeaters is analogous to that of conventional all-optical network.

So, overall, I think there might be some novelty in the way the experiment has been designed (following some underpinning theoretical investigation), but I am not sure if it has been presented to that effect. I believe we are still far away from implementing all-photonic repeaters if we account for what is technologically needed for such systems to offer any advantage. What we need versus what we have got has not been addressed in the paper.

Thank you for stating that there might be some novelty in the way our experiment has been designed (following some underpinning theoretical investigation). We agree with referee's view that the full implementation of all-photonic quantum repeaters is still challenging, as we have already mentioned in the Discussion section. However, it is certain that our experiment proves the working principle of a key ingredient of such all-photonic repeaters. Besides, thanks to your comment, we complemented the theoretical aspect on the potential of our main idea, the combination between time-reversed version of adaptive Bell measurement and the delayed preparation of its resource entanglement. As a result, the revised manuscript is much more informative for readers than the first version and it seems to now clearly show what we need versus what we have got. Thus, we sincerely thank you for presenting us such constructive comments leading to this big improvement of our manuscript.

Dear Referee 2,

Thank you for reviewing our paper and giving us valuable comments. Your constructive comments (coloured blue in what follows) lead to a significant improvement of our paper, as can be confirmed below and by the revised manuscript. In what follows, we would like to reply to your comments in order. Note that the reference numbers appearing below represent ones in the revised manuscript.

In this submission the authors report on the implementation of a key ingredient for all-photon quantum repeaters. The implementation comes together with a new theoretical proposal for Bell measurements. This improvement apparently reduces significantly the resources necessary to implement an all-photon quantum repeater. This is very exciting, since the first proposals for an all-photon quantum repeater seemed out of near future experimental reach or even impractical.

Thank you for stating that our improvement apparently reduces significantly the resources necessary to implement an all-photon quantum repeater and this is very exciting.

However, the paper fails short to its title and abstract. There is no experimental all-photon quantum repeater implemented as the title suggests, and the Bell measurement performed is only half of the newly introduced time-reversed adaptive (TRA) Bell measurement: in the experiment there is only Alice attached to the repeater station. Additionally, I miss a more detailed explanation of this TRA measurement, see comments below. With a more detailed description and accurate claims, I would support the acceptance of this exciting progress towards experimental all-photon quantum repeaters.

Although the presentation of our previous manuscript seemed to have a problem, thank you for supporting the acceptance of our contribution if we provide a more detailed description and accurate claims.

- The title and abstract should be revised to reflect the content. This is a paper about experimental progress towards a key element in all photonic quantum repeaters, the TRA, but with only two parties that are not separated, the experiment cannot be called a repeater.

Thank you for pointing out that the title and abstract should be revised to reflect the content. Following your suggestion, we changed the title and modified the abstract. Please take a look at them.

- Apparently this TRA "needs far fewer single photons than existing all-photon schemes" (page 1). This claim is never substantiated, the authors should either support it with some table, figure, derivation, or drop it. More generally, I miss a comparison with previous proposals in section "All-photon time-reversed adaptive bell measurement"

Thank you for pointing out our inaccurate claim which has existed on page 1. According to your comment, the original sentence, 'In particular, our proposed scheme needs far fewer single photons than existing all-photon schemes', is now replaced by a more accurate claim, 'In particular, our

proposed scheme can be developed by starting from an initial step which requires far fewer single photons than existing all-photonic schemes [21,38]’.

Besides, in the Discussion section, we now give the performance of our protocol through a more detailed calculation, followed by a discussion on how to improve our protocol by replacing the GHZ state with an encoded complete-like cluster state employed in the original all-photonic quantum repeater protocol [21]. In the discussion, we now clarify a reason why our protocol can be performed with far fewer single photons than the original proposal of all-photonic quantum repeaters [21]. The key original idea here is the delayed preparation of multipartite entanglement at the node C. Please take a look at the Discussion section.

You may still wonder why our scheme can be developed by starting from an initial step which requires far fewer single photons than all-photonic intercity QKD [38]. Indeed, the all-photonic intercity QKD protocol is based on the adaptive Bell measurement, similarly to our current proposal. However, in Ref. [38], the adaptive Bell measurement is performed in an active manner with the use of a quantum non-demolition measurement based on an entangled photon source, which is still challenging. This also suggest that it is still challenging to perform even a proof-of-principle experiment of the adaptive Bell measurement, by following the procedure of Ref. [38]. In contrast, our scheme is designed to experimentally confirm the working principle of the adaptive Bell measurement with several photons.

- A minor detail. In section "essence of quantum repeaters" there is a discussion using matter memories, however in the second column of page 2 it is said "by considering the fact that the bell measurement with linear optics succeeds only probabilistically". I am unsure I understood the setup, if we were talking about matter memories, why would we need a linear optics bell measurement?

Thank you for this comment. We replace the original words, ‘By considering the fact that the Bell measurement with linear optics succeeds only probabilistically [37], say with a probability p_s ’ by ‘By including general cases (such as the Duan-Lukin-Cirac-Zoller protocol [18]) where the Bell measurement may succeed only probabilistically, say with a probability p_s ’.

- Second column of page 2 "More precisely, the simplest quantum repeater schemes doubles the communication distance without decreasing the communication efficiency". What does efficiency mean here?

The communication efficiency means an entanglement generation rate. The entanglement generation rate there represents obtained ebits per channel use. To make our argument clearer, we replace ‘More precisely, the simplest quantum repeater scheme doubles the communication distance without decreasing the communication efficiency’ by ‘More precisely, compared with the direct entanglement generation, the simplest quantum repeater scheme enables Alice and Bob to double their communication distance without decreasing the communication efficiency’. Thank you for pointing out a possible confusion from our original sentence.

- In page 7, it is claimed that "teleportation succeeded with high fidelities for the four input states and for any detection events". However, in table 1, one of the entries has fidelity 0.56 +/- 0.11. Two comments: 1) the claim should be moderated to include this entry and 2) I miss some discussion about this anomalous entry in the table.

Thank you for these comments. In photon counting experiments, considering the existence of variations in the observed values, we calculate the standard deviations of the observed fidelities by the observed counts with an assumption of the Poisson statistics. The fidelity 0.56 looks relatively lower than other fidelities, but there are still overlaps with other fidelities within the standard deviation. Thus, we believe that the deviation is caused by the statistical fluctuation. In order to claim clearly, we replace the sentence

" From Table I, we see that teleportation succeeded with high fidelities for the four input states and for any detection events."

by

"Although the fidelities relatively deviate from each other due to the statistical fluctuations caused by the low count events, we can confirm that the teleportation succeeded with high fidelities for most of the four input states and for any detection events."

in page 7.

Dear Referee 3,

Thank you for taking time to read our paper and for giving us helpful comments. Your constructive comments (coloured blue in what follows) lead to a significant improvement of our paper, as can be confirmed below and by the revised manuscript. In what follows, we would like to reply to your comments in order. Note that the reference numbers appearing below represent ones in the revised manuscript.

The manuscript by Hasegawa et al. describes a proof-of-principle experiment for a component of all-photonic quantum repeaters - the time-reversed adaptive Bell state measurement. The research supplements ongoing efforts in all photonic quantum networks and represents experimental research at the forefront of the field. I believe that the article can be of interest to a general science audience and thus support publication in Nature Communications once the following points have been addressed:

Thank you for stating that our article can be of interest to a general science audience and you thus support publication in Nature Communications once we address your points.

-- The results of the proof of concept experiment supersede the classical teleportation fidelity bound by only two s.d. statistical significance, and I believe that a more detailed discussion would be in order. What is the main experimental reason for this limited violation? What are the main challenges/critical technology requirements towards real applications? How does this compare to other approaches for (all-optical) quantum repeaters?

Thank you for these comments. The limitation of the statistical significance is mainly caused by (1) the statistical error due to low count events and (2) the mode mismatch between photons. For (1), we also discussed in our reply to the last comment of referee 2. The mode mismatch (2) is more significant in our experiment. Due to our complex geometry of the experimental setup, the visibility, representing the mode mismatch, was 0.82 at PBS12, which is relatively lower than our typical value 0.90 obtained in a simpler experimental setup [R. Ikuta et al, PRL 106, 110503 (2011)]. We believe this mainly limits the teleportation fidelity in the current experiment and can be improved with integrated photonic circuits and indistinguishable photon sources. To clarify this issue, we add the following sentence at the end of section "PROOF-OF-PRINCIPLE EXPERIMENT".

"The degradation of the process fidelity is mainly caused by the mode mismatch between relevant photons. Especially, it seems that our complex interferometer with GHZ state generation and type-II fusion gates further increases the mode mismatch due to the misalignments. For example, the visibility of 0.82 for two-photon interference at GHZ state generation is relatively lower than typical value 0.90 obtained in a simpler experimental setup [50]. Integrated photonic circuits and indistinguishable photon sources, which have been actively studied recently [25-30, 51], could give a solution to this issue in near future."

The main challenges towards real applications are efficient GHZ photon guns, photon detectors and low loss optical circuits, which have been discussed in ref [37] too. Especially for the photon detectors,

we have discussed qualitatively in the Discussion section of the revised manuscript along with the reviewers' comments. To the best of our knowledge, at this moment, there is no experimental demonstration of adaptive Bell measurement, except for our TRA Bell measurement. Therefore, it is hard to compare our scheme with others in a reasonable way. More general aspect on all-photonic repeater can be seen in refs [21,38].

--In the discussion, the authors state that: "Depending on the efficiency and the size of the GHZ state ... our scheme will gradually expand the achievable distance of the QKD up to 800 km...". The authors should provide quantitative statements in support of this claim, in particular regarding; (i) efficiency; (ii) number of parallel communication channels; (iii) cluster size; (iv) process fidelities, that are required for different link-loss scenarios.

Thank you for giving us this constructive comment. According to this comment, we presented a detailed theoretical analysis on the performance of our protocol in the Discussion section and the Supplementary Note 2, which answers your questions in the sense that we have presented the efficiency of our protocol depending on the local loss for the GHZ state such as the effect of non-unity quantum efficiency of photon detectors. In the calculation, since we have optimized the number of parallel communication channels and its corresponding size of the GHZ state, we have not explicitly mentioned about those. However, as has already been discussed in the section "ESSENCE OF QUANTUM REPEATERS", it is clear that the size is almost in the order of $p_s p_g^{-1} (L/2) \sim (\eta_{L/2})^{-1} \sim e^{L/2 L_{att}}$ when our protocol has a good scaling of $\eta_{L/2}$. Besides, since we have now presented formulas to calculate the performance of our protocol in Supplementary Notes 1 and 2, we think that readers can now reproduce all our calculations. Thank you for giving this comment which has improved our paper.

--While I find the section "essence of quantum repeaters" very illustrative, I believe that the section "All-photonic time-reversed adaptive Bell measurement" should put the new results into clearer context with some of the co-authors previous work (in particular Ref. 19). At present the only link is the reference to the equivalence of the GHZ state in the repeater node to the "...1-st leaf qubits in the original proposal [19]".

Thank you very much for pointing out this. Although we have added the performance of our protocol in Fig. 6, we have noticed that our protocol is quite sensitive to local loss for the GHZ state. Thus, in the Discussion, we have presented an idea on how to overcome this sensitivity. In particular, this idea is to replace the GHZ state with an encoded complete-like cluster state employed in the original all-photonic quantum repeater protocol [21]. Then, we mentioned a striking difference between our protocol and the all-photonic quantum repeaters [21], which stems from our original 'delayed preparation' idea. Therefore, we think that our manuscript makes the difference clearer.

--For someone outside of the field of optical quantum computation, the action of the type-II fusion gate, which plays a key role in the experimental scheme, might be difficult to grasp. I would encourage the authors to add a section describing (in more general terms) the main features, limitations, and its relevance to the overall scheme. This would make the article more accessible to a wider audience, in particular to engineers in the QKD community.

Thank you for this useful recommendation. We now present the detailed description of the type-II fusion gate in Supplementary Note 1 and its connection to the overall scheme.

Also, in the description of the general scheme, the distinction between the polarization qubit and local qubit (Fig 1) seems a little at odds with the "all-photonic" approach. I think this potential source of confusion could be resolved if an intermediate sender node between A(B) & C were mentioned somewhere in the text.

Sorry. We cannot catch your point here. Indeed, the figure 1 wants to show a correspondence between qubits at node C in the matter-qubit-based quantum repeaters and locally prepared polarization qubits at the node C in our all-photonic scheme. If the referee kindly informs us of what is a possible confusion, it is very helpful.

Reviewers' comments:

Reviewer #3 (Remarks to the Author):

The Authors have addressed almost all of the issues pointed out in the previous version of the manuscript. I am thus supportive of publishing the revised manuscript and only have minor comments:

- The Authors discuss the secret key rate for optimal cluster states (plot Fig 6.) in some detail. At some point I think that the approximate size such of optimal cluster states should also be mentioned as it illustrates the technology improvements that are still required for this all-photonic approach to be practical. For example in the Caption of figure 6, the Authors could state the range of optimal cluster states that is spanned (e.g. for 200/400/800 km).

- Regarding my previous comment on Figure 1, the Authors responded: "Indeed, the figure 1 wants to show a correspondence between qubits at node C in the matter-qubit-based quantum repeaters and locally prepared polarization qubits at the node C in our all-photonic scheme. If the referee kindly informs us of what is a possible confusion, it is very helpful." This comment was in reference to the qubits at nodes A(B) in Fig 1(b). I believe it could be a potential source for confusion that the authors explain the working principle of an "all-photonic" quantum repeater using entanglement between a photon and a quantum memory (or at least using the same symbol/nomenclature as in Fig 1a. I understand that the main focus of the article is on node C, but still believe that this could be easily resolved by minor re-wording or a comment in the caption text.

Reviewer #4 (Remarks to the Author):

The manuscript by Hasegawa et al. reports a method and an experimental result that are an important step toward implementing all-optical repeaters. The method uses a large GHZ state generated at the repeater as the initial resource to enable generation of the entanglement with Alice and with Bob. When the photon from Alice or Bob is lost, the corresponding qubit at the repeater should be disentangled. All these entangling and disentangling processes are realized in a single application of the type-II fusion gate. The method itself is quite innovative and the experiment is sound. I think the work is of interest to a general audience and support publication in Nature Communications provided the authors address the following concern:

Going along with the direction of the discussion section which talks about the effect on the key rate when the type-II fusion gate does not properly disentangle unnecessary qubits of the GHZ state, I wonder what the effect on the key rate would be when the locally prepared GHZ state does not have perfect fidelity. When the dimension of the needed GHZ state becomes large, it may be more difficult to generate one with good fidelity.

I was also instructed by the editor to see if the comments of referees 1 and 2 are properly addressed. The discussion section has been revised to discuss about the effect of non-unity efficiency of the photon-number-resolving detectors. I think this adequately addresses the major concern of referee 1 and the other concerns have also been addressed. A major concern of referee 2 about the need for far fewer single photons than existing all-photonic schemes has also been adequately addressed. The other concerns of referee 2 have also been addressed.

Dear referees,

Thank you all for taking time to read our manuscript and our first reply and for giving us your constructive comments. With those, our manuscript is slightly modified.

All the revision parts in this round are marked in the PDF file provided as a Related Manuscript File. If you need, please use it to know which parts are modified in this round.

Dear Referee 3,

Thank you very much for reading our manuscript again. And, your comments throughout the review are very useful for revising our manuscript, which we acknowledge sincerely. In what follows, we reply to your comments (coloured blue in the following).

The Authors have addressed almost all of the issues pointed out in the previous version of the manuscript. I am thus supportive of publishing the revised manuscript and only have minor comments:

Thank you very much for supporting the publication of the revised manuscript.

The Authors discuss the secret key rate for optimal cluster states (plot Fig 6.) in some detail. At some point I think that the approximate size such of optimal cluster states should also be mentioned as it illustrates the technology improvements that are still required for this all-photonic approach to be practical. For example in the Caption of figure 6, the Authors could state the range of optimal cluster states that is spanned (e.g. for 200/400/800 km).

According to this comment, to show the size of the GHZ state, we exemplify the optimal values of the number of multiplexing to obtain Fig. 6 in Supplementary Table I. We hope that this table clarifies how much technology improvements is needed to fully implement our scheme.

- Regarding my previous comment on Figure 1, the Authors responded: "Indeed, the figure 1 wants to show a correspondence between qubits at node C in the matter-qubit-based quantum repeaters and locally prepared polarization qubits at the node C in our all-photonic scheme. If the referee kindly informs us of what is a possible confusion, it is very helpful." This comment was in reference to the qubits at nodes A(B) in Fig 1(b). I believe it could be a potential source for confusion that the authors explain the working principle of an "all-photonic" quantum repeater using entanglement between a photon and a quantum memory (or at least using the same symbol/nomenclature as in Fig 1a. I understand that the main focus of the article is on node C, but still believe that this could be easily resolved by minor re-wording or a comment in the caption text.

Thank you for this clarification. According to this comment, we now introduce matter quantum memories more explicitly in Fig. 1 (a). We also add a comment 'where the qubit a_i (b_i) can be regarded as virtual for the case of application to QKD [21]' to the 2nd paragraph of the section 'Essence of quantum repeaters', in order to clarify the conceptual difference between matter quantum memories at the node C and qubits at nodes A and B. Thank you for pointing out this issue.

Dear Referee 4,

Thank you very much for reviewing not only our manuscript but also our reply in the previous round, very carefully. Your careful review reminds us another thing which we should have mentioned. In this regard, we thank the referee sincerely. In what follows, we reply to your comments (coloured blue in the following).

The manuscript by Hasegawa et al. reports a method and an experimental result that are an important step toward implementing all-optical repeaters. The method uses a large GHZ state generated at the repeater as the initial resource to enable generation of the entanglement with Alice and with Bob. When the photon from Alice or Bob is lost, the corresponding qubit at the repeater should be disentangled. All these entangling and disentangling processes are realized in a single application of the type-II fusion gate. The method itself is quite innovative and the experiment is sound. I think the work is of interest to a general audience and support publication in Nature Communications provided the authors address the following concern:

Thank you for supporting the publication with stating that our method itself is quite innovative, the experiment is sound, and our work is of interest to a general audience.

Going along with the direction of the discussion section which talks about the effect on the key rate when the type-II fusion gate does not properly disentangle unnecessary qubits of the GHZ state, I wonder what the effect on the key rate would be when the locally prepared GHZ state does not have perfect fidelity. When the dimension of the needed GHZ state becomes large, it may be more difficult to generate one with good fidelity.

Thank you for raising a new possible issue on our protocol. In the previous revision, we present the degradation of the efficiency caused by the non-unity quantum efficiency of photon detectors without any modification of our protocol in Fig. 6. Hence, if there is an infidelity of the GHZ state as the referee suggests, the performance of our protocol becomes worse. However, this problem can also be handled by our (previous) proposal that our protocol should be modified to be equipped with the loss-tolerant encoding shown in [21]. Indeed, this loss-tolerant encoding makes our protocol be robust against not only the effect of non-unity quantum efficiency of photon detectors but also depolarization in the transmission channels for photons, as have already been shown in the original paper for all-photon quantum repeaters [21]. Therefore, we think that our protocol is easily modified so as to be robust against a type of the infidelity of the GHZ state. Making our protocol fully fault-tolerant against more general types of infidelity of the GHZ state is an open question in the community of not only all-optical quantum repeaters but also all-optical quantum computation with the loss-tolerant encoding, which seems to be beyond the scope of the present paper.

We finally want to remind that the main focus of this paper is to realize the central part for the adaptive Bell measurement necessitated by all-photon intercity QKD and all-photon quantum repeaters. If the referee acknowledges this part more, we are very grateful.

According to this comment, we have now mentioned this robustness of our protocol against depolarization noise by being equipped with loss-tolerant encoding in the 'Discussion' section. Please look at this modification.

I was also instructed by the editor to see if the comments of referees 1 and 2 are properly addressed. The discussion section has been revised to discuss about the effect of non-unity efficiency of the photon-number-resolving detectors. I think this adequately addresses the major concern of referee 1 and the other concerns have also been addressed. A major concern of referee 2 about the need for far fewer single photons than existing all-photon schemes has also been adequately addressed. The other concerns of referee 2 have also been addressed.

Thank you very much for confirming that our reply and our revision are properly addressed. Your careful reviewing is so helpful even for us.

REVIEWERS' COMMENTS:

Reviewer #4 (Remarks to the Author):

The authors have added that the use of loss-tolerant encoding can make the scheme tolerant to one type of infidelity of the GHZ state, namely the depolarization in the transmission channel for the photons. I believe that this can be done at the expense of making things more messy. I agree that the case of general infidelity is an open problem and that the main focus of the paper is about the adaptive Bell measurement. My concerns have been addressed properly and I recommend publication of this work.